# Circulating SPINT1 Is Reduced in a Preeclamptic Cohort with Co-Existing Fetal Growth Restriction

**DOI:** 10.3390/jcm11040901

**Published:** 2022-02-09

**Authors:** Ciara N. Murphy, Catherine A. Cluver, Susan P. Walker, Emerson Keenan, Roxanne Hastie, Teresa M. MacDonald, Natalie J. Hannan, Fiona C. Brownfoot, Ping Cannon, Stephen Tong, Tu’uhevaha J. Kaitu’u-Lino

**Affiliations:** 1Department of Obstetrics & Gynaecology, Mercy Hospital for Women, The University of Melbourne, Heidelberg, VIC 3084, Australia; spwalker@unimelb.edu.au (S.P.W.); emerson.keenan@unimelb.edu.au (E.K.); hastie.r@unimelb.edu.au (R.H.); teresa.mary.macdonald@gmail.com (T.M.M.); nhannan@unimelb.edu.au (N.J.H.); fiona.brownfoot@gmail.com (F.C.B.); ping.cannon@unimelb.edu.au (P.C.); stong@unimelb.edu.au (S.T.); t.klino@unimelb.edu.au (T.J.K.-L.); 2Mercy Perinatal, Mercy Hospital for Women, Heidelberg, VIC 3084, Australia; cathycluver@hotmail.com; 3Department of Obstetrics & Gynaecology, Stellenbosch University and Tygerberg Hospital, Cape Town 7505, South Africa

**Keywords:** fetal growth restriction (FGR), intra-uterine growth restriction (IUGR), preeclampsia, SPINT1, HAI-1, stillbirth, placental insufficiency

## Abstract

Fetal growth restriction (FGR), when undetected antenatally, is the biggest risk factor for preventable stillbirth. Maternal circulating SPINT1 is reduced in pregnancies, which ultimately deliver small for gestational age (SGA) infants at term (birthweight < 10th centile), compared to appropriate for gestational age (AGA) infants (birthweight ≥ 10th centile). SPINT1 is also reduced in FGR diagnosed before 34 weeks’ gestation. We hypothesised that circulating SPINT1 would be decreased in co-existing preterm preeclampsia and FGR. Plasma SPINT1 was measured in samples obtained from two double-blind, randomised therapeutic trials. In the Preeclampsia Intervention with Esomeprazole trial, circulating SPINT1 was decreased in women with preeclampsia who delivered SGA infants (*n* = 75, median = 18,857 pg/mL, IQR 10,782–29,890 pg/mL, *p* < 0.0001), relative to those delivering AGA (*n* = 22, median = 40,168 pg/mL, IQR 22,342–75,172 pg/mL). This was confirmed in the Preeclampsia Intervention 2 with metformin trial where levels of SPINT1 in maternal circulation were reduced in SGA pregnancies (*n* = 95, median = 57,764 pg/mL, IQR 42,212–91,356 pg/mL, *p* < 0.0001) compared to AGA controls (*n* = 40, median = 107,062 pg/mL, IQR 70,183–176,532 pg/mL). Placental Growth Factor (PlGF) and sFlt-1 were also measured. PlGF was significantly reduced in the SGA pregnancies, while ratios of sFlt-1/SPINT1 and sFlt1/PlGF were significantly increased. This is the first study to demonstrate significantly reduced SPINT1 in co-existing FGR and preeclamptic pregnancies.

## 1. Introduction

Fetal Growth Restriction (FGR), particularly when undetected antenatally, is the single largest risk factor for preventable stillbirth in singleton pregnancies without congenital abnormalities [1]. Despite this, the current means of detecting FGR are inadequate [2], leaving many pregnancies vulnerable to the increased risk of perinatal morbidity and mortality associated with FGR. In many cases, FGR arises due to placental insufficiency, whereby a suboptimal placenta fails to sustain fetal growth. Since placental insufficiency is also implicated in preeclampsia, these pregnancy complications often occur in tandem, as the poorly perfused placenta struggles to maintain the pregnancy with advancing gestation. Identifying biomarkers of placental insufficiency to facilitate improved diagnosis of FGR is therefore of great interest. The current clinical approach is to detect small for gestational age (SGA) fetuses (birthweight < 10th centile), which will likely include most growth-restricted infants and some constitutionally, but not pathologically, small infants.

Serine protease inhibitor, Kunitz type 1 (SPINT1, also known as HAI-1), is a biomarker that we have shown to be deranged in the maternal circulation preceding a diagnosis of SGA at term gestations [3]. Circulating SPINT1 levels were markedly decreased at 36 weeks’ gestation in those women who ultimately delivered an SGA infant, relative to appropriate for gestational age (AGA) counterparts with birthweights >10th centile. Importantly, in addition to being reduced in the maternal circulation, our prior report demonstrated that reduced circulating SPINT1 is associated with key indicators of placental insufficiency. It is most significantly reduced in the circulation of women carrying the smallest babies (birthweight < 3rd centile) and is associated with increased vascular resistance in the uterine artery and reduced placental weight and neonatal body mass index. While marked changes in SPINT1 are apparent close to term gestations, we have also identified reduced circulating levels in pregnancies carrying fetuses destined to deliver SGA infants as early as 24–26 weeks’ gestation [4].

Both preeclampsia and poor fetal growth (indicated by an SGA infant) are believed to originate from placental insufficiency. SPINT1 is a highly expressed placental protein, expressed largely on placental cytotrophoblast cell surfaces [5], which has been demonstrated in murine models to be critical to placentation through its inhibition of its peptide substrates, matriptase and prostasin [6,7]. These results were also recently confirmed in a porcine model [8]. Although our prior work demonstrates reductions in SPINT1 are associated with poor fetal growth, no studies have found its expression to be deranged in preeclampsia.

It was therefore hypothesised that SPINT1 perturbations associated with FGR might also be observed in those with a primary diagnosis of preeclampsia. If confirmed, it means that it could be a biomarker of FGR even among women with preeclampsia. At the biological level it would imply FGR in preeclampsia has a distinct molecular pathogenesis to preeclampsia without FGR. In this study, we therefore measured circulating SPINT1 protein levels in women with pregnancies complicated by preterm preeclampsia, to assess whether SPINT1 was differentially expressed among those with and without an SGA infant at birth.

## 2. Materials and Methods

Blood samples from two clinical trials testing novel therapeutics for preeclampsia, provided a unique opportunity to examine SPINT1 in women with concomitant preeclampsia and FGR, as indicated by delivery of a small for gestational age (SGA) infant. Both trials included women with early-onset preeclampsia who had varying degrees of disease severity. As such, this study was able to assess whether, in the context of placental insufficiency manifesting as preterm preeclampsia, SPINT1 is still decreased in pregnancies complicated by growth restriction as previously established. Both trials assessed prolongation of pregnancy following randomisation and collected maternal plasma samples at randomisation and twice weekly until delivery.

### 2.1. PIE Trial

The Preeclampsia Intervention with Esomeprazole (PIE) trial was a randomised, double-blind, placebo-controlled clinical trial assessing esomeprazole as a treatment of preterm preeclampsia in women undergoing expectant management. Women with preterm preeclampsia, diagnosed between 26 + 0– and 31 + 6–weeks’ gestation, were randomised to a daily 40 mg dose of either esomeprazole or placebo. Specific eligibility and exclusion criteria have previously been described in detail [9,10]. Participant characteristics are detailed in Appendix A. This study had ethical approval from Stellenbosch University Health Research Ethics Committee (M14/09/038, Federal Wide Assurance Number (FWAN) 00001372, Institutional Review Board (IRB) number 0005239).

### 2.2. PI-2 Trial

The Preeclampsia Intervention 2 (PI-2) trial was a randomised, double-blind, placebo-controlled trial of metformin to treat preterm preeclampsia in women undergoing expectant management. Women with preterm preeclampsia, diagnosed between 26 + 0– and 31 + 6–weeks’ gestation, were randomised to 3 g of either metformin or an identical placebo, administered in divided daily doses. Specific eligibility and exclusion criteria have previously been described in detail [11,12]. Patient characteristics are detailed in Appendix A. The trial had ethical approval from Stellenbosch University Health Research Ethics Committee (M16/09/037, FWAN00001372, IRB0005239).

### 2.3. Classification of Samples

Plasma samples were classified as either Appropriate for Gestational Age (AGA) or SGA according to customised infant birthweight centile above or below the 10th centile (respectively), determined post-partum using the Gestation-Related Optimal Weight (GROW) Bulk Centile calculator (v8.0.4, 2019).

### 2.4. Measurement of Plasma Protein Levels

SPINT1 protein levels in maternal plasma samples were ascertained using a SPINT1 ELISA kit (Sigma-Aldrich, St. Louis, MO, USA), following the manufacturer’s specifications. A dilution of 1:5 was deemed optimal and used for both cohorts. Plasma placental growth factor (PlGF) and soluble FMS-like tyrosine kinase-1 (sFlt-1) was measured using a commercial electrochemiluminescence immunoassay platform (Roche Diagnostics).

### 2.5. Statistical Analysis

Statistical analyses were performed using GraphPad Prism 9 (GraphPad Software, Inc., San Diego, CA, USA), with all data first assessed for Gaussian distribution and subsequently analysed using appropriate statistical tests. No data points were excluded as outliers. Maternal characteristics and birth-outcome data (Appendix A) of FGR pregnancies relative to AGA “controls” were compared using Mann–Whitney U, unpaired t-, Fisher’s exact, or Chi square tests. For linear regression analysis, the natural logarithm of each biomarker value (lnSPINT1 or lnPlGF) was used for fitting. For all biomarker data, when two groups were compared, a Mann–Whiney U-test was used. For more than two groups (i.e., >10th v. 3rd–10th v. <3rd), a Kruskal–Wallis test was used and post-hoc analyses ascertained by Dunn’s multiple comparisons test.

## 3. Results

### 3.1. Circulating SPINT1 Is Reduced in Women with Preterm Preeclampsia Complicated by SGA–PIE Trial

SPINT1 levels were first measured in all samples to determine the effect of esomeprazole treatment on the concentration of circulating SPINT1 (Figure 1a). Compared to placebo-treated participants, women in the esomeprazole group did not have significantly altered SPINT1 levels. Therefore, we utilised all available samples from trial participants (*n* = 97 of a total 120 participants), regardless of the treatment received, for further analyses of SPINT1, sFlt-1, and PlGF. Having selected the samples collected closest to delivery, the median gestation at sampling across all samples was 29 + 3 weeks (Appendix A). In preeclampsia cases who delivered an SGA infant, the gestation at delivery and the interval between sampling and delivery were significantly decreased with a median value of 219 days (IQR 204–230 days, *p* < 0.01) and 14 days (IQR 6–21 days, *p* < 0.05), respectively, compared to 236 days (IQR 215–239 days) and 20 days (IQR 12–33 days) in preeclamptic patients delivering an AGA infant (Appendix A).

In the PIE cohort, SPINT1 levels were significantly decreased in preeclamptic patients who subsequently delivered an SGA infant (Figure 1b, *n* = 75) with a median SPINT1 concentration of 18,857 pg/mL (IQR 10,782–29,890 pg/mL, *p* < 0.0001), relative to preeclamptic participants who delivered an AGA infant (*n* = 22, median = 40,168 pg/mL, IQR 22,342–75,172 pg/mL). Our previous study confirmed that circulating SPINT1 at 36 weeks’ gestation was most deranged in women with infants who ultimately had a birthweight <3rd centile [3], so we next divided the <10th centile group into those <3rd centile (*n* = 59) and those between 3rd and 10th centiles (*n* = 16). In doing so, we found that the change in SPINT1 associated with SGA pregnancies was driven by the most severe cases of growth restriction (those SGA infants with a birthweight <3rd centile). In these severe instances of SGA (Figure 1c), SPINT1 was significantly reduced with a median SPINT1 concentration of 14,516 pg/mL (IQR 9967–27,627 pg/mL, *p* < 0.0001) relative to the AGA group (median = 40,168 pg/mL, IQR 22,342–75,172 pg/mL), The SGA pregnancies delivering between 3rd and 10th birthweight centiles, however, were not accompanied by any significant changes in SPINT1 (median = 26,416 pg/mL, IQR 17,911–36,975 pg/mL, *p* = 0.2). The relationship between lnSPINT1 expression in maternal plasma and birthweight centile was then assessed (Figure 1d), and a significant association was identified using linear regression (r^2^ = 0.1117, *p* = 0.0008).

Having established these differences in SPINT1 levels, we next assessed Placental Growth Factor (PlGF) in the same samples (Figure 1e). As expected, PlGF levels were significantly reduced, in both <3rd (median = 27.94 pg/mL, IQR 15.65–40.67 pg/mL, *p* < 0.0001) and 3rd–10th samples (median = 39.47 pg/mL, IQR 20.42–93.51 pg/mL, *p* = 0.01) relative to AGA controls (median = 121.7 pg/mL, IQR 57.67–214.1 pg/mL). The association between PlGF and birthweight centile (Figure 1f) was also significant (r^2^ = 0.2424, *p* < 0.0001).

### 3.2. Validation That SPINT1 Is Reduced in Women with Preterm Preeclampsia Complicated by SGA–PI-2 Trial

Given that circulating SPINT1 was significantly reduced in SGA pregnancies in PIE, we next sought to validate this observation in the PI-2 cohort. We initially confirmed that metformin did not significantly alter circulating SPINT1 relative to the placebo-treated cohort (Figure 2a). This again allowed us to combine data from all available samples for further analyses (*n* = 135 of 180 participants), with a median gestation of 34 + 1 weeks (AGA) and 32 + 2 weeks (SGA; Appendix A). In preeclampsia cases who delivered an SGA infant, the gestation at delivery and interval between sampling and delivery was significantly decreased, with a median value of 227 days (IQR 212–239 days, *p* < 0.0001) and 5 days (IQR 3–7 days, *p* < 0.001), respectively, compared to 239 days (IQR 237–240 days) and 8 days (IQR 6–11 days) in preeclamptic patients delivering an AGA infant (Appendix A).

As observed in PIE, SPINT1 was significantly reduced in the plasma of preeclamptic pregnancies complicated by SGA (*n* = 95, median = 57,764 pg/mL, IQR 42,212–91,356 pg/mL, *p* < 0.0001), relative to those who delivered an AGA infant (*n* = 40, median = 107,062 pg/mL, IQR 70,183–176,532 pg/mL, Figure 2b). Analysis of this second cohort reflected the findings of PIE, with significantly decreased SPINT1 levels in only the most severely growth-restricted pregnancies (<3rd birthweight centile; *n* = 77, median = 54,871 pg/mL, IQR 42,037–78,771 pg/mL, Figure 2c, *p* < 0.0001) compared to AGA controls (median = 107,062 pg/mL, IQR 70,183–176,532 pg/mL), whereas there was no significant decrease in those pregnancies delivering an infant in the 3rd–10th birthweight centile (*n* = 18, median = 84,542 pg/mL, IQR 41,280–118,013 pg/mL, *p* = 0.1). A similar association between lnSPINT1 levels and birthweight centile, as seen in PIE, was also observed in PI-2 (Figure 2d, r^2^ = 0.1520, *p* < 0.0001). Absolute differences in SPINT1 between cohorts likely reflect differences in assay kit batches associated with research-grade ELISAs. Importantly, each cohort was run using ELISAs from the same manufactured batch.

Again, we also measured PlGF concentrations in the PI-2 samples. We found that relative to AGA controls (median = 321.2 pg/mL, IQR 97.6–550.4 pg/mL), there was a reduction in PlGF concentration in the 3rd–10th cohort (*n* = 18, median = 35.9 pg/mL, IQR 17.4–228.0 pg/mL, *p* = 0.001) and the <3rd cohort (*n* = 77, median = 29.7 pg/mL, IQR 21.4–47.1 pg/mL, *p* < 0.0001). The association of lnPlGF concentration with birthweight centile (Figure 2f) was also verified (r^2^ = 0.3822, *p* < 0.0001).

### 3.3. A Ratio of sFlt-1/SPINT1 to Identify Placental Insufficiency Manifesting as SGA Co-Exdisting with Preterm Preeclampsia

sFlt-1 is an anti-angiogenic molecule that is highly elevated in preeclampsia [13]. A ratio of sFlt-1/PlGF offers an excellent rule-out test for patients who might develop preeclampsia in the coming weeks as it has a high negative predictive value [14]. We assessed whether sFlt-1/SPINT1 might differ in preterm preeclamptic pregnancies delivered SGA. In PIE, the sFlt-1/SPINT1 ratio (Figure 3a) showed significant elevations in those patients who ultimately delivered SGA, at both <3rd (*n* = 59, median = 1.4, IQR 0.44–2.54, *p* < 0.0001) and 3rd–10th (*n* = 16, median = 0.82, IQR 0.28–1.39, *p* = 0.01) birthweight centiles, relative to AGA controls (*n* = 22, median = 0.09, IQR 0.05–0.60). The most significant changes were observed in the <3rd birthweight centile cohort. These results were mirrored by the sFlt-1/PlGF ratio (Figure 3b; <3rd median = 655, IQR 456–1426, *p* < 0.0001; 3rd–10th median = 524, IQR 98.9–1191, *p* = 0.006; control median = 37.3, IQR 11.5–278); however, the sFlt-1/PlGF ratio demonstrated greater differentiation in the median values between 3rd–10th centile samples and controls (*p* = 0.006) than the sFlt-1/SPINT1 ratio (*p* = 0.01).

The analyses of the PI-2 study confirmed these changes, with the sFlt-1/SPINT1 ratio (Figure 3c) being elevated in both <3rd (*n* = 77, median = 0.27, IQR 0.17–0.48, *p* < 0.0001) and 3rd–10th (*n* = 18, median = 0.06, IQR 0.04–0.20, *p* = 0.04) birthweight centile groups, compared to AGA controls (*n* = 40, median = 0.02, IQR 0.01–0.05). The sFlt-1/PlGF ratio (Figure 3d) was similarly increased, relative to AGA controls (median = 7.45, IQR 3.58–40.43), in 3rd–10th (median = 316.3, IQR 9.59–1208, *p* = 0.001) and <3rd (median = 503.1, IQR 302.8–910.4, *p* < 0.0001) groups. Again, there was greater differentiation between 3rd–10th centile samples and controls using the sFlt-1/PlGF ratio (*p* = 0.001), compared to the sFlt-1/SPINT1 ratio *p* = 0.04); although both ratios demonstrated significant elevations in both severe and moderate SGA (indicated by a birthweight centile <3rd and 3rd–10th, respectively).

## 4. Discussion

Preterm FGR, though less common than term FGR, is associated with a poorer prognosis [15] and often co-exists with preeclampsia. In the search for biomarkers for placental insufficiency, we identified SPINT1 as a promising candidate. This study explored the potential that SPINT1 might be associated with preterm FGR in the presence of coexisting preeclampsia (around 30 weeks’ gestation). Indeed, we found in two separate cohorts that SPINT1 was markedly reduced in the plasma of patients with preeclampsia delivering SGA infants and preeclampsia, relative to those women with preeclampsia carrying appropriately grown infants. As expected, PlGF was particularly reduced in the pregnancies complicated by both preeclampsia and FGR, while a ratio of sFlt-1/SPINT1 was significantly elevated in a similar manner to that observed for sFlt-1/PlGF.

We previously reported significantly reduced SPINT1 concentrations in the circulation of normotensive pregnancies complicated by preterm FGR [3], however this is the first time SPINT1 has been examined in association with preterm preeclampsia. Decreased circulating SPINT1 was found to be strongly associated with severe growth restriction (birthweight <3rd centile) within preeclamptic pregnancies. While PlGF performs very strongly in this cohort with established disease, further analyses in an unselected cohort collected before disease onset are needed to compare the true predictive capacity of both proteins in preterm disease. Certainly, our data in an unselected cohort at 36 weeks’ gestation suggests that SPINT may perform better than PlGF in SGA prediction at term [3].

In both cohorts, it was found that the gestation at delivery and the interval between sampling and delivery was reduced in preeclamptic pregnancies delivering an SGA infant compared to AGA controls (Appendix A). This earlier delivery may relate to clinical actions that were taken in respect to suspected SGA or due to physiological factors in the pregnancy itself. Future work is required to specifically investigate whether there are changes in the predictive capability of each biomarker in cases where clinical action was taken due to suspected SGA or due to underlying maternal co-morbidities.

Our finding that SPINT1 is differentially expressed in pregnancies complicated by both preeclampsia and FGR provides further understanding of the pathogenesis of these disorders. SPINT1 is a protein highly expressed in the placenta, and our prior work demonstrates an association of decreased circulating levels with reduced placental weight [3]. Though both preeclampsia and FGR are considered to be disorders of placental insufficiency, our data herein suggest some differences in placental mechanisms underlying the manifestation of preeclampsia in the presence or absence of FGR. Animal [7,8,16] and human [3,17] placental studies provide strong evidence that SPINT1 is involved in placental development. Whether there are alterations in placental development in preeclampsia complicated by FGR, relative to preeclamptic placentas that facilitate normal growth, has not been carefully studied and should be the focus of ongoing work.

In this study, there were more cases of coexisting preeclampsia and SGA (FGR) than those with preeclampsia uncomplicated by SGA (Appendix A). This is because preeclampsia and FGR tend to concurrently arise at preterm gestations due to early placental insufficiency. FGR is less common as a comorbidity of preeclampsia diagnosed at term, which highlights the importance of carefully selecting biomarkers that could stratify those at risk during preterm gestations.

There remains a dearth of biomarkers that can accurately predict preterm or term SGA or FGR, which is an ongoing challenge within the field. Current clinical tools, such as maternal and fetal risk factors, or ultrasound measures, perform modestly [1,18]. SPINT1 is a novel biomarker that we recently reported as being differentially expressed in pregnancies destined to deliver small for gestational age at term gestations, performing better than PlGF in that setting. To date, universal ultrasound in the third trimester in nulliparouns women provides the highest sensitivity for prediction of term SGA, sitting at 57% compared to 20% for selective, or clinically indicated ultrasound [19]. Thirty-six-week SPINT1 performs similarly to selective ultrasound for predicting SGA, however its utility if combined with other biomarkers or clinical measures remains the focus of ongoing work. This new study provides the first insights into SPINT1 possibly being deranged in preterm growth restriction, irrespective of co-existing preeclampsia. Validation of this finding, however, would require a very large prospective collection, which might also include ultrasound measures so that a direct comparison between these clinical tools could be accurately assessed.

## 5. Conclusions

This study demonstrates that SPINT1 levels are decreased in preterm pregnancies with placental insufficiency manifesting as FGR, even in the presence of concurrent preeclampsia. It further validates the association between low SPINT1 and poor fetal growth we have previously reported [3]. In addition, this work highlights the potential of SPINT1 as a novel biomarker that could be added to the toolbox for preterm disease, but further studies are required to confirm its predictive performance.

## Figures and Tables

**Figure 1 jcm-11-00901-f001:**
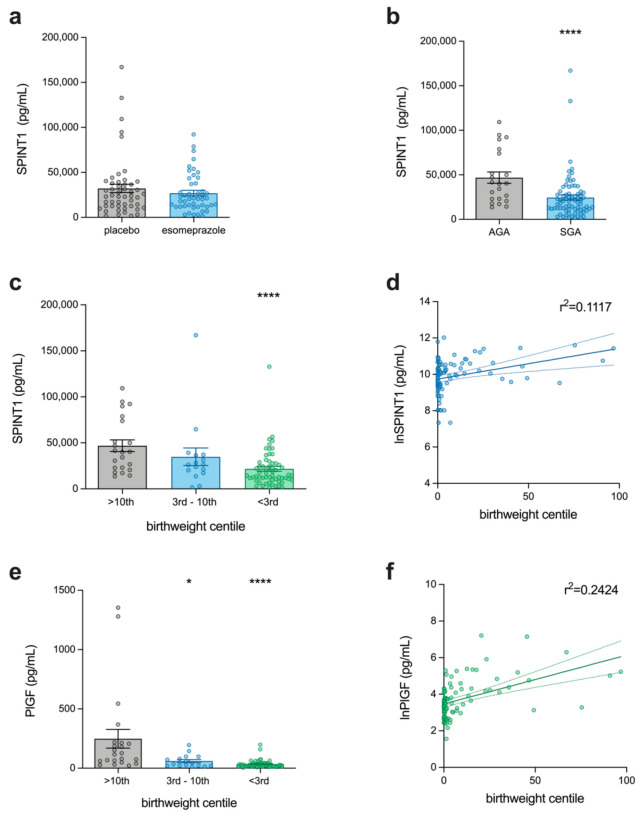
Circulating SPINT1 levels relative to infant birthweight centile in preeclamptic patients (PIE trial samples). (**a**) Esomeprazole intervention (*n* = 47) had no significant influence (*p* = 0.63) on SPINT1 levels compared to placebo (*n* = 50); (**b**) SPINT1 levels were decreased in small for gestational age (SGA) pregnancies (*n* = 75), relative to appropriate for gestational age (AGA) controls (*n* = 22); (**c**) this difference was most marked in severe SGA (birthweight <3rd centile, *n* = 59), with no significant decrease in moderately SGA pregnancies (birthweight centile 3rd–10th, *n* = 16) relative to AGA controls (birthweight ≥ 10th centile, *n* = 22); (**d**) lnSPINT1 levels also increased with higher birthweight centile (r^2^ = 0.1117, *p* = 0.0008); (**e**) PlGF levels were significantly decreased in severe (*n* = 59) and moderately (*n* = 16) SGA pregnancies, again most markedly in severe cases, relative to AGA controls (*n* = 22); (**f**) lnPlGF levels increased with birthweight centile (r^2^ = 0.2424, *p* < 0.0001). Each data point represents a single patient; *n* = 97 total, * *p* < 0.05, **** *p* < 0.0001.

**Figure 2 jcm-11-00901-f002:**
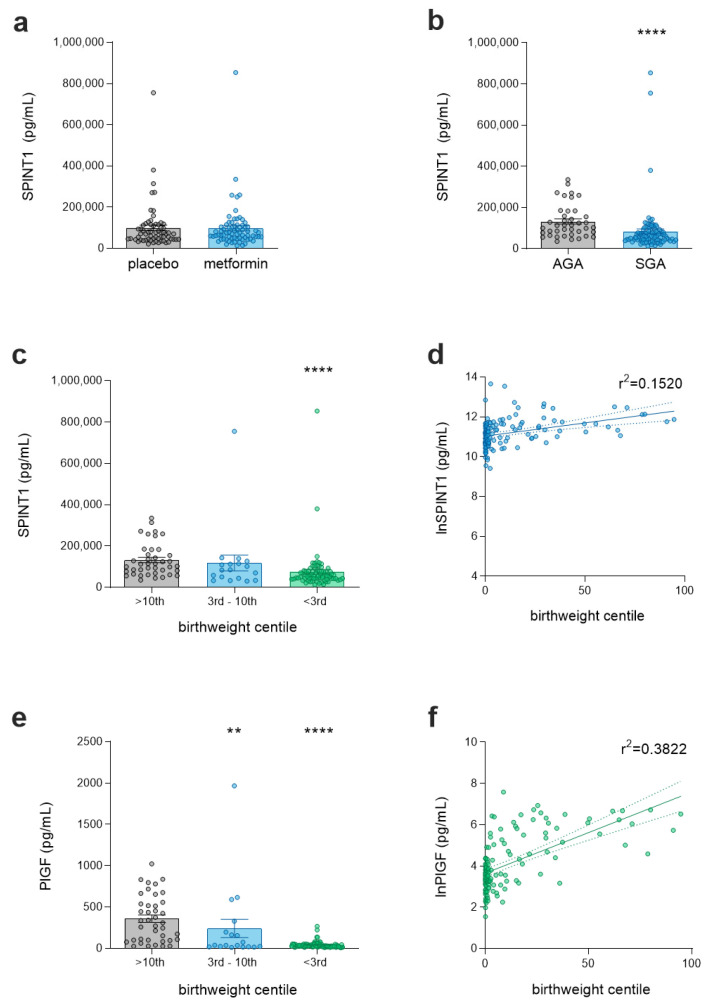
Validation of circulating SPINT1 levels relative to infant birthweight centile in preeclamptic patients (PI-2 trial samples). (**a**) Metformin intervention (*n* = 72) had no significant influence (*p* = 0.63) on SPINT1 levels compared to placebo (*n* = 63); (**b**) SPINT1 levels were decreased in SGA pregnancies (*n* = 95), relative to AGA (birthweight ≥ 10th centile) controls (*n* = 40); (**c**) this difference was most marked in severe SGA (birthweight <3rd centile, *n* = 77), with no significant decrease in moderately SGA pregnancies (birthweight centile 3rd–10th, *n* = 18) relative to controls (*n* = 40); (**d**) lnSPINT1 levels were increased with higher birthweight centile (r^2^ = 0.1520, *p* < 0.0001); (**e**) PlGF levels were significantly decreased in severe (*n* = 77) and moderately *(**n* = 18) SGA pregnancies, again most markedly in severe cases, relative to AGA controls (*n* = 40); (**f**) lnPlGF levels increased with birthweight centile (r^2^ = 0.3822, *p* < 0.0001). Each data point represents a single patient; *n* = 135 total, ** *p* < 0.01, **** *p* < 0.0001.

**Figure 3 jcm-11-00901-f003:**
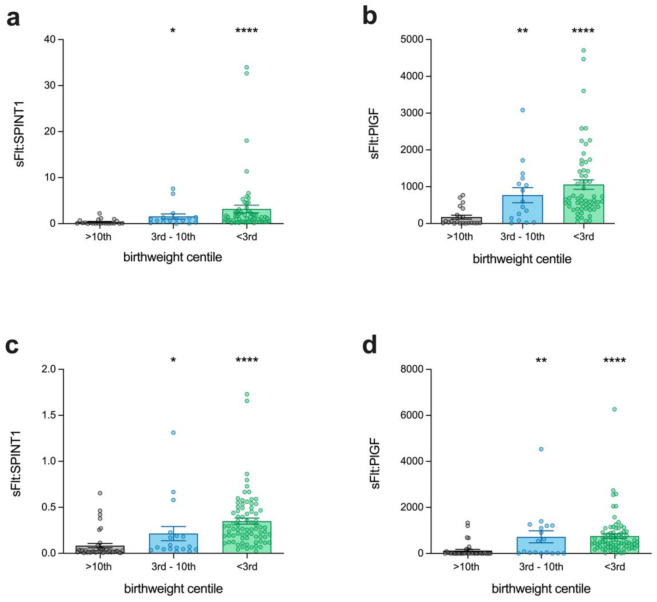
Association between sFlt-1 ratio and birthweight centile when substituting SPINT1 for PlGF. (**a**) The ratio of sFlt-1 to SPINT1 in PIE samples increased stepwise with decreasing birthweight centiles (<3rd *n* = 59; 3rd–10th *n* = 16; >10th *n* = 22); (**b**) the existing ratio of sFlt-1 to PlGF was also elevated with decreasing birthweight centile; (**c**) the sFlt-1 and SPINT1 ratio was again increased stepwise with SGA severity in the PI-2 samples (<3rd *n* = 77; 3rd–10th *n* = 18; >10th *n* = 40); (**d**) the sFlt-1 and PlGF ratio was again also elevated in SGA. Each data point represents a single patient; * *p* < 0.05, ** *p* < 0.01, **** *p* < 0.0001.

## Data Availability

Raw data are available upon request to corresponding author.

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
