# Peer review of "Circulating SPINT1 Is Reduced in a Preeclamptic Cohort with Co-Existing Fetal Growth Restriction"

_jcm, 2022, doi:10.3390/jcm11040901_

Round 1
Reviewer 1 Report
This manuscript is well written.
However, its interest is limited.
The authors do not really argue about the utility of their study.
Angiogenic factors (PlGF and sFLT1 above all) are proven reliable biomarkers of complications due to placental insufficiency (in particular pre-eclampsia). PlGF can be used as early as the first trimester of pregnancy to detect preterm pre-eclampsia and sFLT1/PlGF ratio is used to rule out other diagnoses than pre-eclampsia and has a prognostic value when measured iteratively after pre-eclampsia diagnosis. Its interest in IUGR's diagnosis or prediction is less obvious.
However, this study does not highlight a real additive value of SPINT1 in the diagnosis of IUGR associated with pre-eclampsia, in comparison with PlGF or with sFlt1/PLGF.
It does not show either if SPINT1 value is able to predict IUGR associated with pre-eclampsia before it is detectable on ultrasonography, or if the changes in SPINT1 values over time have a prognostic value (for example, could predict stillbirth).
It does not compare either SPINT1 values to doppler findings.
Last, but not least, the authors did not compare the SPINT1 values measured in pathological pregnancies to healthy pregnancies, which is the biggest limit of this study.
The discussion is very short and the authors have to develop current knowledge about SPINT1 (fundamental data in particular), as it is the main topic of the manuscript.
Reviewer 2 Report
The manuscript evaluated the relationship between a novel biomarker and birthweights in two selected preeclamptic cohorts. This exploratory study came to the conclusion that SPINT1, a protease inhibitor/ Serine peptidase inhibitor, was reduced in those pregnancy where the birthweight was below the 10th centile and correlated with the birthweight.
The manuscript is well structured and was a pleasure to read. The biggest criticism of this manuscript is that there is no control group to help in the interpretation of these results. If the authors can add some control data if would greatly improve the validity of these results. Overall, the topic is very relevant, and deserves to be eventually published.
Specific comments:
- Introduction:
- It will help the reader if the authors give some detail of the expressivity of SPINT1, since it is not specific to syncytiotrophoblasts or cytotrophoblast.
- Methodology:
- Although the two cohorts are clearly defined with the time point of samplings, the specific timing of the last sampling with relation to the delivery is not clear. Could the authors please include both the gestation at delivery for both cohorts as well as the interval between sampling and delivery.
- The biomarker was only interpreted in the context of birthweight, but this marker could also have been influenced by many other variables. Could the authors please include either the incidences of maternal morbidity or the reasons for eventual delivery to understand the context of the clinical condition at last sample.
- Results:
- Please state the why certain samples were excluded i.e., in the PIE cohort there were 97 samples of 120 and in the PI-2 cohort only 135 of 180.
- Figures:
- Please state the actual number of data points for each graph.
- Did the authors test for outliers i.e., two-sided Grubbs' test? And where any data points excluded on this basis?
- The discussion could benefit from:
- The brief discussion could be greatly improved by highlighting why SPINT1 specifically could breach this gap in FGR, how this biomarker directly compares to other established criteria from ultrasound to PlGF. What are the benefits of this marker?
- This marker has been reviewed in other clinical contexts but has suffered from low sensitivity, is this a foreseeable problem in the context of FGR screening. Especially with the wide ranges and different absolute values seen between the cohorts.
Author Response
Reviewer #2
The manuscript is well structured and was a pleasure to read. The biggest criticism of this manuscript is that there is no control group to help in the interpretation of these results. If the authors can add some control data if would greatly improve the validity of these results. Overall, the topic is very relevant, and deserves to be eventually published.
Thank you. Please see the comments in response to reviewer #1 which address the lack of controls here.
Specific comments:
- Introduction:
- It will help the reader if the authors give some detail of the expressivity of SPINT1, since it is not specific to syncytiotrophoblasts or cytotrophoblast.
This is a good point; we have amended as follows:
SPINT1 is a highly expressed placental protein, expressed largely on placental cytotrophoblast cell surfaces [5], which has been demonstrated in murine models to be critical to placentation through its inhibition of its peptide substrates, matriptase and prostasin [6, 7].
- Methodology:
- Although the two cohorts are clearly defined with the time point of samplings, the specific timing of the last sampling with relation to the delivery is not clear. Could the authors please include both the gestation at delivery for both cohorts as well as the interval between sampling and delivery.
We thank the reviewer for raising this point. We have updated Supplementary Table 1 and 2 to include the gestation at delivery and interval between sampling and delivery in both cohorts. These results have been added to the main manuscript on Pages 3 and 5 indicating that:
- In the PIE cohort, in preeclampsia cases who delivered an SGA infant the gestation at delivery and interval between sampling and delivery were significantly decreased with a median value of 219 days (IQR 204-230 days, p <0.01) and 14 days (IQR 6-21 days, p<0.05) respectively, compared to 236 days (IQR 215-239 days) and 20 days (IQR 12-33 days) in preeclamptic patients delivering an AGA infant.
- In the PI-2 cohort, in preeclampsia cases who delivered an SGA infant the gestation at delivery and interval between sampling and delivery were significantly decreased with a median value of 227 days (IQR 212-239 days, p <0.0001) and 5 days (IQR 3-7 days, p<0.001) respectively, compared to 239 days (IQR 237-240 days) and 8 days (IQR 6-11 days) in preeclamptic patients delivering an AGA infant.
We have added a discussion of the impact of these findings on Page 8 of the revised manuscript as follows:
“In both cohorts, it was found that the gestation at delivery and the interval between sampling and delivery was reduced in preeclamptic pregnancies delivering an SGA infant compared to AGA controls. This earlier delivery may relate to clinical actions that were taken in respect to suspected SGA, or due to physiological factors in the pregnancy itself. Future work is required to specifically investigate whether there are changes in the predictive capability of each biomarker in cases where clinical action was taken due to suspected SGA or due to underlying maternal co-morbidities.”
- The biomarker was only interpreted in the context of birthweight, but this marker could also have been influenced by many other variables. Could the authors please include either the incidences of maternal morbidity or the reasons for eventual delivery to understand the context of the clinical condition at last sample.
Unfortunately, we do not have access to information on the incidence of maternal morbidities or reasons for eventual delivery in our current dataset. We however believe this is an important area of discussion and have highlighted its relevance for future works in the updated discussion on Page 8 as below:
“Future work is required to specifically investigate whether there are changes in the predictive capability of each biomarker in cases where clinical action was taken due to suspected SGA or due to underlying maternal co-morbidities.”
- Results:
- Please state the why certain samples were excluded i.e., in the PIE cohort there were 97 samples of 120 and in the PI-2 cohort only 135 of 180.
Since there were varying numbers of blood collections between participants, and plasma samples from these cohorts have been used for other biomarker screening as part of the clinical trial publications, not all samples were available for analysis of all three biomarkers (SPINT1, sFlt, PlGF). As such, only the samples which had readings for all three biomarkers were included, hence the lower numbers of samples.
- Figures:
- Please state the actual number of data points for each graph.
Thanks for picking up on this; we have included all the relevant n numbers in Figure legends, particularly where SGA was divided into <3rd and 3rd–10th centiles.
- Did the authors test for outliers i.e., two-sided Grubbs' test? And where any data points excluded on this basis?
We thank the reviewer for highlighting this point. No outliers were excluded in the current work based on previous findings which have shown that a small number of biological samples occasionally return large biomarker readings on both research-grade and commercial assay platforms [1]. We are confident that as rank based statistical methods such as the Mann-Whitney and Kruskal-Wallis tests were used for all two- or three-group biomarker comparisons, the statistical results obtained should be robust against small numbers of samples departing from the typical biomarker range.
However, we agree that the influence of these large biomarker values should be addressed when performing linear regression analysis. As such, we have updated the regression analyses presented in Figures 1d, 2d and 2f, and in the associated sections of the manuscript (on Pages 3-6) to use log-transformed biomarker readings lnSPINT1 and lnPlGF as the dependent regression variable. All statistical associations as identified in the original manuscript remain significant after log transformation.
- The discussion could benefit from:
- The brief discussion could be greatly improved by highlighting why SPINT1 specifically could breach this gap in FGR, how this biomarker directly compares to other established criteria from ultrasound to PlGF. What are the benefits of this marker?
- This marker has been reviewed in other clinical contexts but has suffered from low sensitivity, is this a foreseeable problem in the context of FGR screening. Especially with the wide ranges and different absolute values seen between the cohorts.
- We thank the reviewer for these thoughts, and have now included a paragraph in the discussion to try to address these points. We note that in the current study we did not have universal ultrasound data to compare SPINT1 levels with, but have indeed made comparsions with PlGF, which we have discussed.
Reviewer 3 Report
Authors' studies circulating SPINT1 levels in pregnant women and found that they would be decreased in co-existing preterm preeclampsia and FGR.
Author Response
Reviewer #3
Authors' studies circulating SPINT1 levels in pregnant women and found that they would be decreased in co-existing preterm preeclampsia and FGR.
We thank the reviewer for this summary, and note that there are no questions requiring response.
References
- Kaitu'u-Lino, T.J., et al., Circulating SPINT1 is a biomarker of pregnancies with poor placental function and fetal growth restriction. Nat Commun, 2020. 11(1): p. 2411.
- Kaitu'u-Lino, T.J., et al., Maternal circulating SPINT1 is reduced in small-for-gestational age pregnancies at 26 weeks: Growing up in Singapore towards health outcomes (GUSTO) cohort study. Placenta, 2021. 110: p. 24-28.
- Haragan, A.F., et al., Diagnostic accuracy of fundal height and handheld ultrasound-measured abdominal circumference to screen for fetal growth abnormalities. Am J Obstet Gynecol, 2015. 212(6): p. 820.e1-8.
Round 2
Reviewer 1 Report
Thank you for this revised version of your manuscript.
Questions and commentaries of the reviewers have been addressed and the revised version of the manuscript is fine.
Reviewer 2 Report
The authors have addressed most of the concerns in the manuscript.
Please ensure all graphs state the actual number of data points fig 1a, 1e, 2a, 2e.
Author Response
We are grateful for the opportunity to improve upon our manuscript “Circulating SPINT1 is reduced in a preeclamptic cohort with co-existing fetal growth restriction” with the help of the reviewers’ feedback. We have amended the manuscript accordingly, with minor adjustments in the second round of review, namely the inclusion of all data point sums in the legends for Figures 1a, 1e, 2a and 2e (highlighted), as per reviewer #2’s request.